# Application and Optimization of the Thin Electric Heater in Micro-Injection Mold for Micropillars

**DOI:** 10.3390/nano12101751

**Published:** 2022-05-20

**Authors:** Can Weng, Qianfan Tang, Jiangwei Li, Lintao Nie, Zhanyu Zhai

**Affiliations:** 1State Key Laboratory of High Performance and Complex Manufacturing, Central South University, Changsha 410083, China; canweng@csu.edu.cn (C.W.); 17804332312@163.com (Q.T.); lijiangwei@csu.edu.cn (J.L.); 203711062@csu.edu.cn (L.N.); 2College of Mechanical and Electrical Engineering, Central South University, Changsha 410083, China

**Keywords:** micro-injection molding, thin electric heater, variotherm system, micropillar, PP

## Abstract

The development of variotherm systems has helped to improve the quality of micro-injection molded products. Thin electric heaters have significant advantages in size, efficiency and installation convenience. However, the use of thin electric heaters has brought the problem of non-uniform temperature distribution of the insert. The good replication of the functional surfaces containing microstructures quickly and uniformly is still a challenge. In this work, the heating performance of the thin electric heater in a variotherm system is investigated by combining numerical simulations with experiments. Micro-injection molding of PP micropillars was also performed. The obtained results show that the addition of a transition layer with high thermal conductivity in the heating structure can optimize the uniformity of the temperature distribution of the insert. Furthermore, the replication heights of the micropillars can be significantly increased by the developed variotherm mold, which provides a new idea for the optimal design of a local variotherm system.

## 1. Introduction

Components containing microstructures have been widely used in biomedical [1], aerospace [2,3], scientific and technological electronics [4,5] and other industries. However, it is difficult to produce these components in batch by conventional manufacturing technologies. Micro-injection molding technology, which originated in the 1980s, is widely used to produce polymer parts with complex geometries and various sizes by virtue of its low cost, high efficiency and flexibility.

In contrast to conventional injection molding, the rheological properties and thermodynamic behavior of polymers during the micro-injection molding process are usually influenced by scale effects [6]. As a result, many physical phenomena are very different from the conventional injection molding process, and the influences of each processing parameter on the molding quality are different [7].

In conventional injection molding, a variotherm system based on rapid mold heating and cooling is often used to improve the surface quality of products. In micro-injection molding, the goal of the application of the variotherm system is to improve the quality of microstructures. The molding defects that may occur in conventional injection molding may not exist in micro-injection molding. However, bending, fracture, stretching, necking, surface burr and incomplete filling of microstructures become the main molding defects [8].

The mold temperature and the cavity pressure have significant impacts on the molding quality of micro-injection molded products [9,10]. G. Lucchetta and M. Sorgato from the University of Padua, Italy, experimentally investigated the technical limitations of PS material for micro-injection molding for microstructures with a high aspect ratio. They indicated that, among these main processing parameters, the mold temperature had the greatest effect on the average height of the microstructures.

When the cavity temperature increased from 60 to 100 °C, the average molding height of the microstructures increased from 0.37 to 1.14 μm. At the same time, the uneven distribution of the molding pressure was an important factor that led to different regions with different heights of microstructures [11]. In addition, many scholars have mentioned that the optimal cavity temperature should be close to or higher than the glass transition temperature of the polymer [12].

A cavity temperature that is overly low will cause the premature formation of a frozen layer of the polymer melt [13,14]. In that case, the melt is prevented from entering the cavity, resulting in incomplete replication of the microstructures. Therefore, variotherm systems are always used to help the melt fill into micron- and sub-micron-sized cavities and to adapt the internal morphology for better performance [15]. Furthermore, it cannot be neglected that non-uniform temperature distribution of the insert containing microstructures will seriously affect the overall molding quality of the microstructures, due to the temperature sensitivity of the forming of microstructures. The heating and cooling processes are important sources of non-uniform temperature distribution of the cavity.

To shorten the molding cycle and improve the product quality, some scholars have used infrared heating [16,17] or induction heating [18,19] to optimize the temperature distribution. However, traditional electric heating and water cooling are still the most widely used rapid heating methods in injection molding. To improve the competitiveness of electric heating, Takushi [20] from the University of Tokyo, Japan, investigated near-electric heating and used it to fabricate microscale structures. He demonstrated that thin electric heaters consume less heat during the rapid heating and cooling of the mold. He also found that, under the conditions of close heating, there was a certain temperature gradient on the same insert, which inevitably affected the replication quality of microstructures at different locations on the insert.

Wang and Zhao [21] from Shandong University, China, suggested in their study that the average heating rate of the cavity increased as the distance between the cavity and the heat source decreased. The high temperature rise generated by close heating would inevitably lead to a non-uniform distribution of the insert temperature. In summary, the advantages and disadvantages of close heating are clear. The heating rate is sufficiently fast, and the energy utilization is very high. However, this also brings the problem of non-uniform temperature distribution of the insert.

Considering the problems in these studies, an improved method of the variotherm system based on close heating is proposed in this work. We analyzed the factors affecting the temperature uniformity of the insert. A transition layer with high thermal conductivity is added between the close heating source and the insert to control the heat transfer path. The temperature distribution of the insert is optimized with a guaranteed heating rate. An injection mold for micro-pillars is designed and manufactured. The surface temperature distribution of the insert is measured and evaluated by simulations and experiments. Finally, the developed mold is used to fabricate PP products with micro-pillars, and the effectiveness of the improved method in optimizing the insert temperature distribution is verified.

## 2. Rapid Mold Heating and Cooling Method

The Rapid Heat Cycle Mold (RHCM) developed in this study is based on wafer resistance heating and the water cooling on the insert block as shown in Figure 1. Circulating water cooling was used as the cooling method, and was positioned close to the surface of the mold cavity. This arrangement increases the cooling efficiency. The developed mold had a transition layer with high thermal conductivity between the heat source and the cavity, compared to a conventional electrically heated rapid thermal cycle mold. The transition layer was added to provide a uniform temperature distribution in the mold cavity. The high thermal conductivity would be helpful to quickly obtain heat from the point heat source and to avoid overheating in certain areas of the insert.

The thin electric heater used in this paper is also known as a thick film resistor. The thin electric heater was made of an insulating layer, resistance paste, protective glazes and other materials (as shown in Figure 2), which were sintered at a high temperature on the substrate by screen printing technology, allowing the cavity temperature to be raised at a faster rate. The overall dimensions of both the thin electric heater and the insert containing microstructures were 25 × 25 × 1 mm. The thickness of the thin electric heater was 1 mm, which increases the heating efficiency, simplifies the installation and removal of the electric heater and compacts the mold.

## 3. Evaluation of Thermal Response by Numerical Simulation

### 3.1. Analytical Model

The 3D transient heat transfer models were built by COMSOL Multiphysics 5.6. The RHCM model and the meshing of the cavity used in FEM are shown in Figure 3. A physically controlled mesh was used.

Natural air convection was set on the mold surfaces. The initial value of the overall mold temperature was 20 °C—the same as the ambient temperature. The temperature of the cooling water was 15 °C. A boundary heat source was used to simulate the heating process of the thin electric heater. The thermal conductivity (λ), heat capacity (C) and density (ρ) of the mold are shown in Table 1.

### 3.2. Three Heating Modes

Three heating modes with a thin electric heater were compared by simulation. The three modes were direct heating, the addition of a flat graphite transition layer and the addition of a graphite transition layer with a circular hole. The heating power of the thin electric heater was 120 W/cm^2^.

#### 3.2.1. Direct Heating

Figure 4 shows the temperature distribution of the insert when heated directly using a thin electric heater. The heating time was 2.7 s when the average temperature of the insert reached 373 K (equivalent to 100 °C). The highest temperature was located in the center of the insert at approximately 399–403 K (equivalent to 126–130 °C). The edges of the insert had the temperatures of 331–352 K (equivalent to 58–79 °C).

Temperatures were obtained from the five positions of the insert. The sampling locations are shown in Figure 5. The measured temperature curves for these five points as a function of time are shown in Figure 6.

The results show that, when using a thin electric heater with a high temperature uniformity, the temperature distribution of the insert still exhibited significant non-uniformity. For an insert with a side length of 20 mm, the temperature difference between the center and the edge was 50–60 °C. The occurrence of this situation will have a negative impact on the overall molding quality of microstructures.

#### 3.2.2. The Addition of a Flat Graphite Transition Layer

The material of the transition layer was graphite, and the thermal conductivity was set to 150 W/m°C. The graphite layer was located between the heat source and the insert and had dimensions of 25 × 25 × 1 mm. Figure 7 shows the temperature distribution of the insert with the addition of a graphite transition layer. When the average temperature of the insert reached 100 °C, the heating time was 3.4 s. The highest temperature was concentrated in the center of the insert at approximately 392–398 K (equivalent to 119–125 °C). The edges of the insert had temperatures of 340–354 K (equivalent to 67–81 °C).

Figure 8 illustrates the temperature variations at five points on the insert. It can be seen that the maximum and minimum values of temperature on the insert decreased to varying degrees, and the maximum values decreased greatly. The non-uniform temperature distribution of the inserts also improved slightly but to a lesser extent. There is still a significant difference between the temperature in the center of the insert and the temperature at the edges. Thus, we will explore this issue further.

#### 3.2.3. Addition of a Graphite Transition Layer with a Circular Hole

In order to avoid the excessive heating in the center of the insert-containing microstructures, a hole structure was added to the center of the transition layer to avoid direct contact between the insert and the transition layer. The solid heat transfer process between them was isolated, and only a relatively small amount of air convection heat transfer took place to reduce the concentration of heat in the center. The convective heat transfer coefficient was set to 30 W/m^2^ in the simulation model.

Figure 9 shows the structure of the transition layer containing a circular hole with a diameter of 8 mm and a depth of 0.3 mm. Under the same boundary conditions, the temperature distribution of the insert is shown in Figure 10.

The simulation results in Figure 11 show that, when the average temperature of the insert reached 100 °C, the heating time was 3.7 s. The highest temperature was concentrated at the point where the insert contacted with the edge of the circular hole, which was approximately 380–385 K (equivalent to 107–112 °C). The temperature in the center of the insert was 385 K (equivalent to 112 °C), and its edges were 346–363 K (equivalent to 73–90 °C). At the same time, the temperature uniformity in the central part of the insert was better.

The standard deviation (SD) is introduced to evaluate the temperature distribution uniformity of the inserts. The standard deviations of the heating temperature for the three heating modes are compared, and the results are shown in Figure 12. It can be seen that the SD of the temperature distribution tends to be stable as the heating time increases.

When the distance between the heat source and the insert is close and the direct heating mode is used, the uniformity of the temperature distribution of the insert is the worst. By adding a transition layer with high thermal conductivity between the heat source and the insert, the uniformity of temperature distribution can be improved. At the same time, the uniformity of the temperature distribution of the insert can be further improved by machining a hole in the transition layer in the temperature concentration area of the insert.

### 3.3. Effect of the Transition Layer on Mold Heating

#### 3.3.1. Diameter of the Circular Hole

To further understand the effect of the circular hole size on the uniformity of the temperature distribution of the insert, the effects of transition layers with circular hole diameters of 4, 5, 6, 7 and 8 mm on the temperature distribution and uniformity of the insert are compared in Figure 13.

When the average temperature of the insert reaches 373 K, the central peak value decreases significantly with increasing diameter of the insulated hole, and the proportion of the temperature uniform part of the center in insert increases considerably as shown in Figure 14. With the decrease of SD, the temperature distribution on the surface of the insert becomes more uniform. When there is no transition layer, the average temperature of the insert reaches 373 K within 2.7 s. When a transition layer without a circular hole is added, the heating time is 2.9 s.

When a transition layer with an 8 mm circular hole is added, the heating time is 3.2 s. There is small time difference among these cases. However, when the radius changes to 10 mm, the heating time in the simulation analysis increases significantly to 4.5 s. This is because the actual contact area between the transition layer and the insert is significantly reduced due to the large diameter of the round hole, thereby, resulting in a slow temperature rise rate in the overall temperature of the insert. The size of the circular hole has a direct influence on the temperature distribution of the insert. In practice, the size and the number of circular holes should, therefore, be determined by the size and the shape of the insert.

#### 3.3.2. Thermal Conductivity of the Transition Layer

In addition, materials with different thermal conductivities for the transition layer with an 8 mm circular hole have been compared. Common materials with high thermal conductivity, such as graphite and copper, were mainly considered. The thermal conductivity of graphite was set to 150 W/(m°C), and that of copper was set to 380 W/(m°C). The standard deviation of the temperature of the insert with these two transition layers is compared in Figure 15.

As the thermal conductivity of the transition layer material increases, the temperature of the insert is more uniformly distributed. As the heating time increases, the standard deviation of the temperature of the insert increases continuously in the absence of the transition layer. When the graphite transition layer is added, the standard deviation of the insert temperature is stabilized at 20–22.5.

When the copper transition layer is used, the standard deviation of the insert temperature is stabilized at 12.5–15.0, which is an obvious improvement. At the same time, the heating time increases from 3.2 to 3.5 s when the thermal conductivity of the transition layer increases from 150 to 380 W/m°C. This is due to the fact that, when the thermal conductivity of the transition layer increases, the heat exchange between the transition layer and the contact part of the mold outside the insert also increases. Then, the heat energy obtained by the insert decreases, which leads to a longer heating time.

## 4. Experiments

### 4.1. Material and Equipment

The evolution of the insert temperature distribution during the RHCM micro-injection molding process was investigated to further demonstrate the feasibility of temperature optimization. Taking the micropillars as an example, the insert temperature measurement and micro-injection molding experiments were performed. The machine used in the experiments was an all-electric injection molding machine provided by Sodick Company (Yokohama, Japan, model: LD05EH2).

The molding material was PP (HD-120MO). To improve the filling quality of the micropillars, the average surface temperature of the insert was heated to 100 °C. The cooling stage should be below 50 °C to allow the parts to solidify completely and to avoid microstructure deformation caused by high temperature during the demolding. Using the developed mold (as shown in Figure 16), the RHCM test system was established.

### 4.2. Thermal Response Measurements of the Insert

Thermal response measurement experiments of the insert surface were conducted using a developed mold. The temperature data were monitored and recorded in real time by five k-type temperature sensors fixed on the surface of the insert. The heater was a thin electric heater with a total thickness of 1 mm, a rated voltage of 24 V and a heating power of 120 W. According to the simulation results, the thermal responses of the insert surface under three heating modes were compared in the experiments. The three modes were directing heating, the addition of a graphite transition layer containing an 8 mm hole and the addition of a pure copper transition layer containing an 8 mm hole.

### 4.3. Micro-Injection Molding of Microstructures

To investigate the effect of the improved method on the molding of microstructures, micro-injection molding experiments with micropillars were performed. Inserts with microstructures were fabricated by lithographic silicon masters and precision Ni electroforming [22]. The microstructures formed a regular array of quadrilateral prisms as shown in Figure 17. The average width of the microholes on the nickel mold insert was 23.3 μm, and the average depth was 60.2 μm.

The injection molding experiments of micropillars were also performed under three heating modes as described above for comparison. The molding temperature in the center of the insert was controlled at 100 °C, and the circulating medium temperature was set to 50 °C. LSM was used to examine the morphology of the molded PP parts. The average height of five micropillars in an area centered on the sampling point were measured, and the results were processed by statistical analysis.

## 5. Results and Discussion

### 5.1. Thermal Responses of Inserts

Figure 18 illustrates the curves of temperature variation with time at five sampling points on the surface of the insert under the three heating modes. Figure 19 compares the mean square errors of the variation of surface temperature of the insert with time for the three modes.

The experimental results are in general agreement with the simulation results. For the whole insert, the temperature was higher at the center and lower at the edges. When the temperature of the insert tended to be stable, the maximum temperature difference under direct heating was 39.3 °C. When using a graphite transition layer with an 8 mm hole, the maximum temperature difference was 23.8 °C.

When a copper transition layer with an 8 mm hole was used, the temperature difference was 20.1 °C. The temperature difference between point C and point B at the edge of the insert decreased from 22.2 to 10.7 and 9.7 °C under the modes of direct heating and having transition layers. The addition of a transition layer with high thermal conductivity resulted in a more uniform temperature distribution of the insert, and the temperature uniformity was improved by nearly 50%.

The graphite transition layer plays a similar role to the copper transition layer in improving the temperature distribution of the insert. Considering the temperature uniformity, the copper transition layer with a hole was slightly better than the graphite transition layer with a hole. However, the copper transition layer takes longer to heat up to the set temperature. This is because the thermal conductivity of pure copper is greater than that of graphite. The faster the rate of thermal conductivity of the copper transition layer is, the more uniform the temperature distribution is formed. At the same time, the heat transferred from the copper transition layer to the mold is relatively small, and thus the temperature rises more slowly.

In the case of close heating, the insert will obtain a more uniform temperature distribution when the heating rate of the heat source is small. If the heat source heats faster, the temperature will rise faster. When the thermal conductivity of the mold steel is low, there is a large temperature gradient. At this time, the heat transfer rate between the molds is small, and it is difficult to meet the requirements of uniform temperature distribution of the inserts. Therefore, adding a highly thermally conductive transition layer with a circular hole between the heat source and the insert is a good solution.

The high thermal conductivity of the transition layer ensures the rate of heat transfer, which rapidly transfers the heat from the point heat source to the lower temperature area. The circular hole structure in the middle isolates the solid contact heat transfer between the transition layer and the insert, while the convective heat transfer of air in the circular hole is negligible. The air acts as an insulation layer, avoiding the concentration of heat in the center of the insert. Both simulations and experiments showed that this method ensures good temperature uniformity of the insert when using closerange heating.

Compared with the simulation, the average temperatures of the insert increased more slowly in the experiment. The gap between the heat source layer, the transition layer and the insert was the primary reason.

### 5.2. Molding of Microstructures

The molding quality of micropillars, especially the height of molded micropillars, can be an important indicator for evaluating this proposed method. The replication qualities of injection molded micropillars with three heating modes were compared.

The mold temperature rose until the stabilization for the molding process was at about 15 s and until the packing time was 10 s. With the help of the mold temperature control machine, the drop from the molding temperature to the demolding temperature of 50–60 °C was approximately 12 s. Micropillars at different sampling positions on the surface of injection molded PP products were examined by LSM. The molding heights of the micropillars at each sampling point are shown in Figure 18, where ‘a’ presents the average thickness of the substrate and ‘b’ presents the average height of five micropillars.

From Figure 20, it can be seen that, under the three heating modes, the molding heights of the micropillars near the center point C did not differ much—approximately 55.76–56.70 μm—while the molding heights of points A, B, D and E have large differences. When there was no transition layer, the average heights of point A, B, D and E were 28.98, 41.32, 45.20 and 49.45 μm, respectively. When a graphite transition layer with a hole was added, the forming height of the micropillars in each sampling region was improved. In particular, the height in region A increased to 32.25 μm, and that in region E increased to 52.78 μm.

We demonstrated that the insert temperature uniformity had a great effect on the molding height of the microstructures. When a copper transition layer with a hole was added, the average heights of point A, B, D and E were 37.21, 49.55, 49.14 and 54.30 μm, respectively, with average height increases of 28.4%, 20.0%, 8.8% and 9.8%, respectively. The experimental results proved that the addition of a transition layer with a hole could effectively improve the molding heights of the microstructures.

The maximum molding height of PP micropillars was 56.70 μm, which did not reach the average depth of 60.2 μm of the microholes in the Ni mold insert. The main reason may be that the air in the cavity causes the melt to be unable to fill the microholes completely. The cavity evacuation can be tested to solve this problem in future research. From the experimental results, the temperature uniformity of the mold insert is one of the important factors affecting the molding height of the micropillars. The replication quality of the micropillars is sensitive to the temperature of the insert.

A high insert temperature is beneficial to the filling of PP microstructures; however, the resulting high molding temperature can easily lead to the deformation of PP microstructures, which can seriously affect the replication quality. Therefore, a high molding temperature and a low demolding temperature are necessary for the molding of microstructures. The heat transfer path between the heat source and the insert is blocked by the circular hole that does not contact the insert, and there is only a slight convective heat transfer between the two.

The temperature increase in the center of the insert mainly depends on the other areas in contact with the transition layer. This method avoids excessive heat concentration in the center of the insert and effectively reduces its temperature rise rate. The other areas obtain more heat from the heat source, especially the edge areas, which results in a more uniform temperature distribution of the insert, thus, improving the overall molding height of the micropillars.

## 6. Conclusions

In the present work, the performance of a thin electric heater in a variotherm system was evaluated, and the temperature uniformity of the mold insert was improved by adding a transition layer. The advantages and disadvantages of the thin electric heater in micro injection molding were further illustrated in the case of forming micropillars. The main conclusions can be drawn as follows:

(1) A thin electric heater with local heating can realize the rapid temperature change of the insert over a wide range, and the method of combining the thin electric heater with a transition layer is a good solution to improve the temperature uniformity of the insert.

(2) The thermal conductivity of the transition layer and the size and number of holes have an important effect on the temperature distribution of the insert. The increase of the thermal conductivity of the transition layer and the diameter of the circular hole can effectively improve the temperature distribution uniformity of the insert.

(3) In the experiment, the injection molding cycle of the micropillars could be controlled within 40 s. The thin electric heater not only improved the efficiency but also greatly increased the energy utilization. Temperature had a significant effect on the forming of microstructures. By adding a transition layer, the height of micropillars in the edge region of the insert was increased from 28.98–45.20 to 37.21–49.55 μm.

(4) There was some difference between the height of the molded micropillar and the actual cavity depth of 60.2 μm. The cavity pressure would be the main reason for this difference. Regarding the molding quality of the micropillars in each area, microstructures should not be designed within 1–2 mm of the edges of the insert.

In general, reasonable rapid thermal circulation is important and necessary for micro-injection molding. Adding a transition layer with a circular hole allows the heat transfer path of the heater to be controlled and the temperature field in the cavity to be optimized. Based on this method, the “Conformal Heating” technique can be further developed.

## Figures and Tables

**Figure 1 nanomaterials-12-01751-f001:**
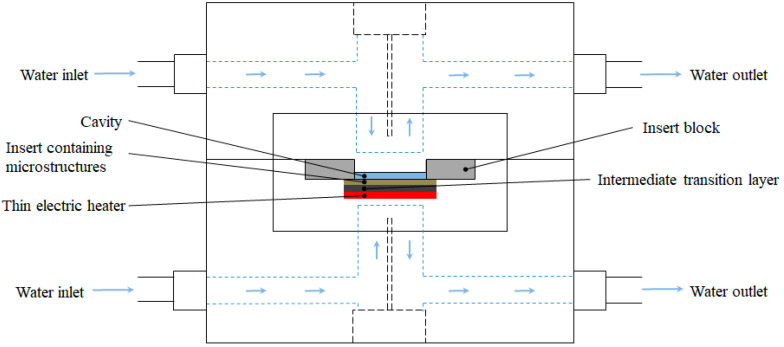
Schematic diagram of the developed rapid heat cycle mold.

**Figure 2 nanomaterials-12-01751-f002:**
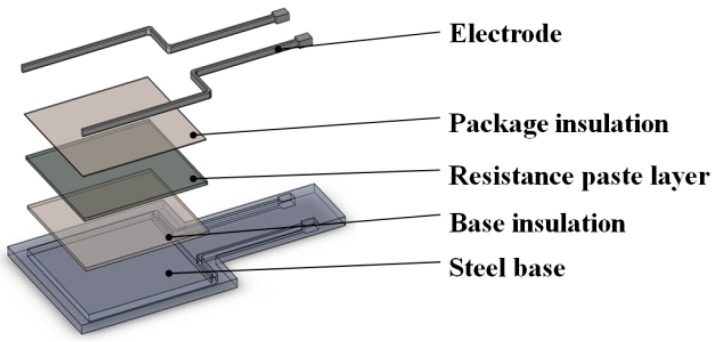
Structure diagram of the thin electric heater.

**Figure 3 nanomaterials-12-01751-f003:**
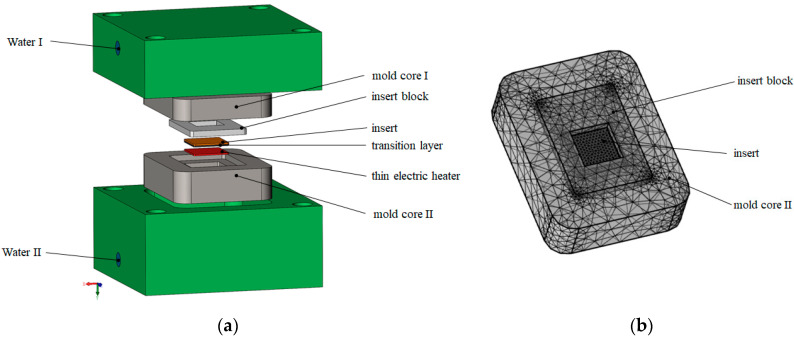
(**a**) Heat transfer model and (**b**) meshing of the cavity.

**Figure 4 nanomaterials-12-01751-f004:**
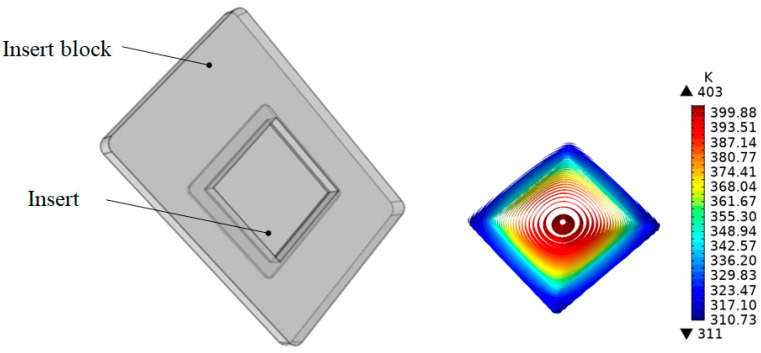
The temperature distribution of the insert with direct heating.

**Figure 5 nanomaterials-12-01751-f005:**
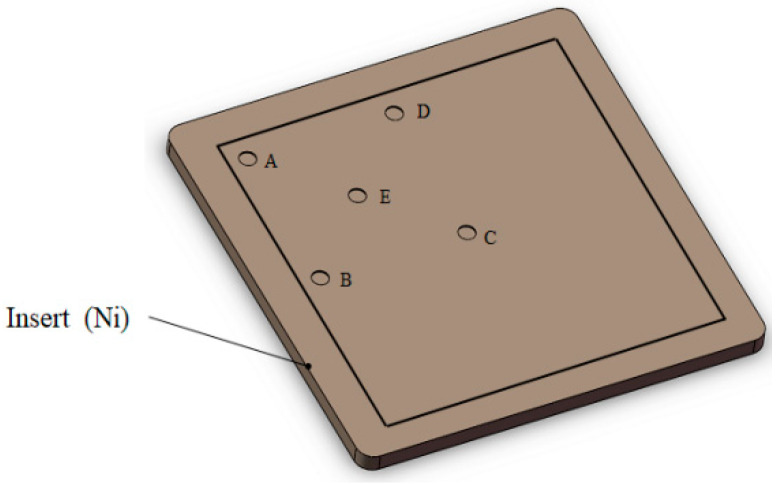
Temperature measurement sampling points.

**Figure 6 nanomaterials-12-01751-f006:**
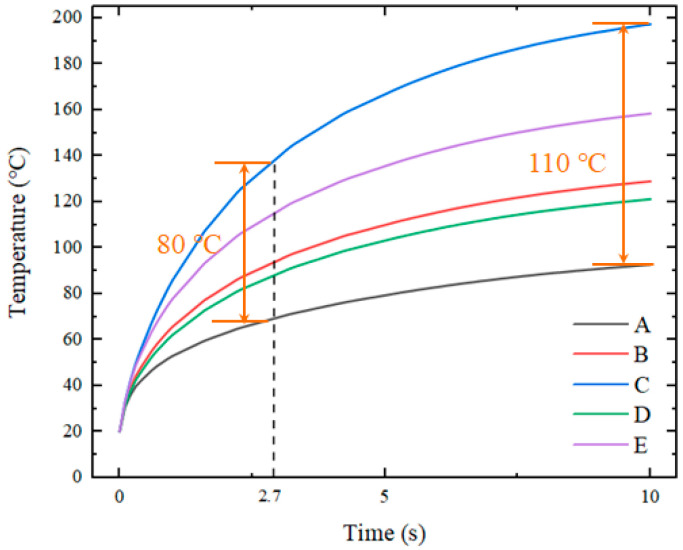
Curves of temperature versus time for five points with direct heating.

**Figure 7 nanomaterials-12-01751-f007:**
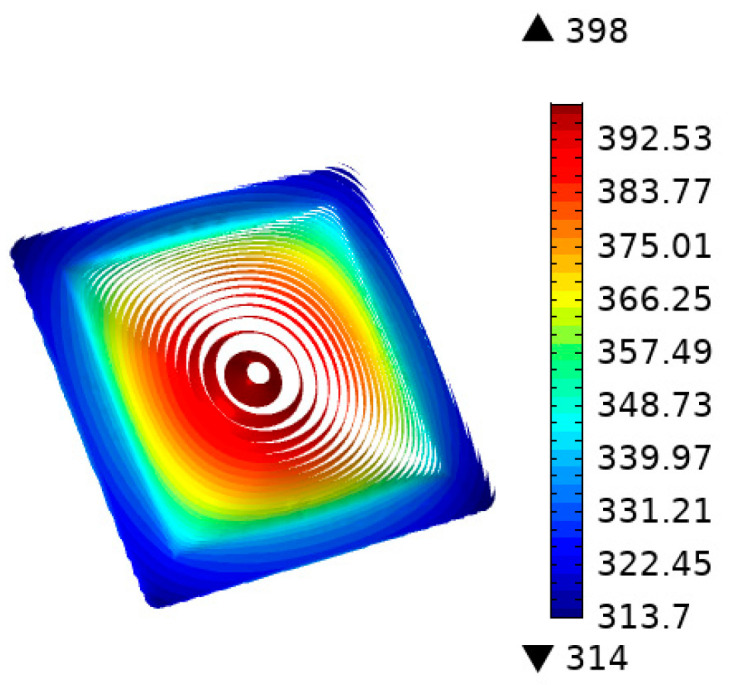
Temperature distribution of the insert with a flat graphite layer.

**Figure 8 nanomaterials-12-01751-f008:**
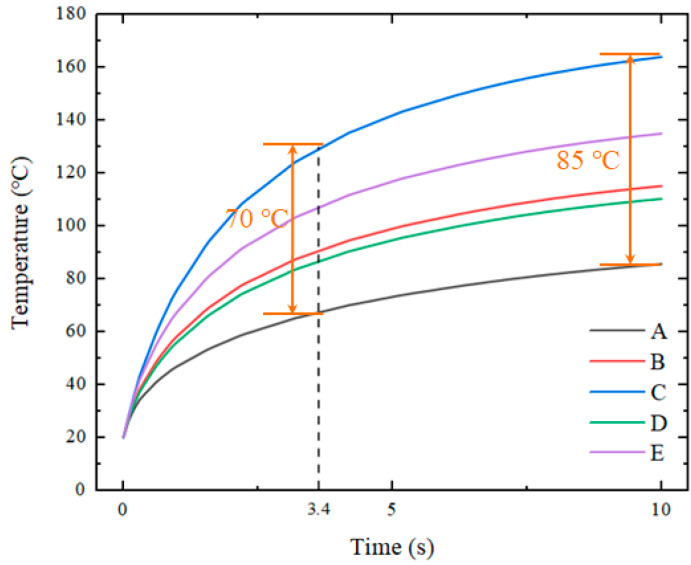
Temperature response of five points with the addition of a flat graphite layer.

**Figure 9 nanomaterials-12-01751-f009:**
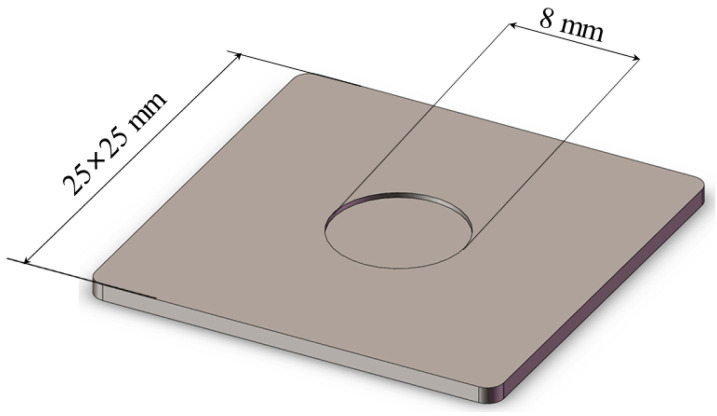
Transition layer structure with an 8 mm circular hole.

**Figure 10 nanomaterials-12-01751-f010:**
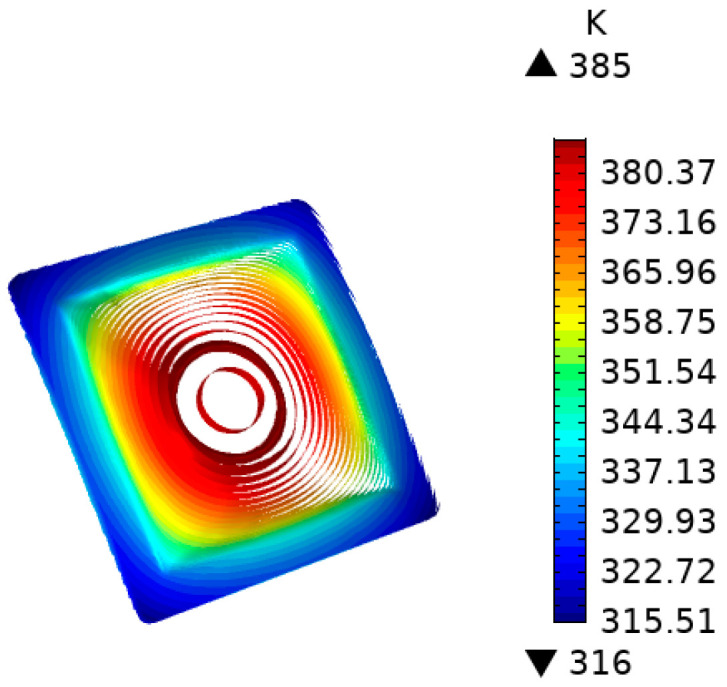
Temperature distribution of insert adding a transition layer with an 8 mm circular hole.

**Figure 11 nanomaterials-12-01751-f011:**
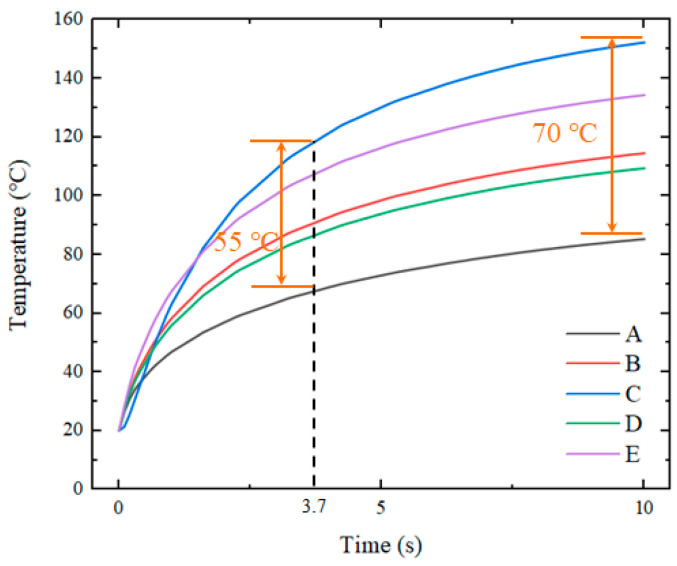
Temperature response of the insert with a graphite transition layer containing an 8 mm hole at five sampling points.

**Figure 12 nanomaterials-12-01751-f012:**
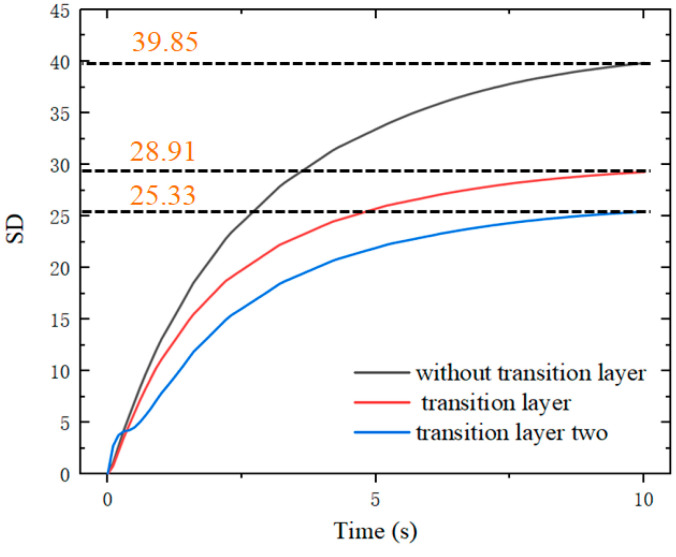
Temperature standard deviations for the three heating modes.

**Figure 13 nanomaterials-12-01751-f013:**
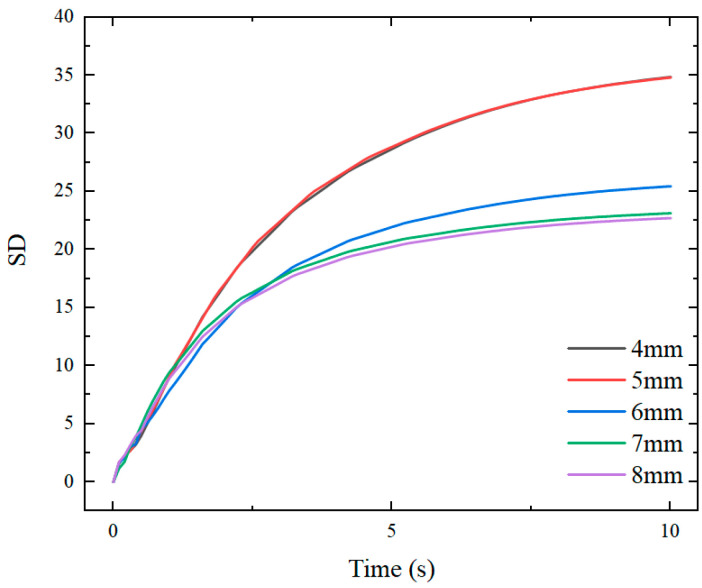
SD of the insert temperature with transition layers under different circular hole diameters.

**Figure 14 nanomaterials-12-01751-f014:**
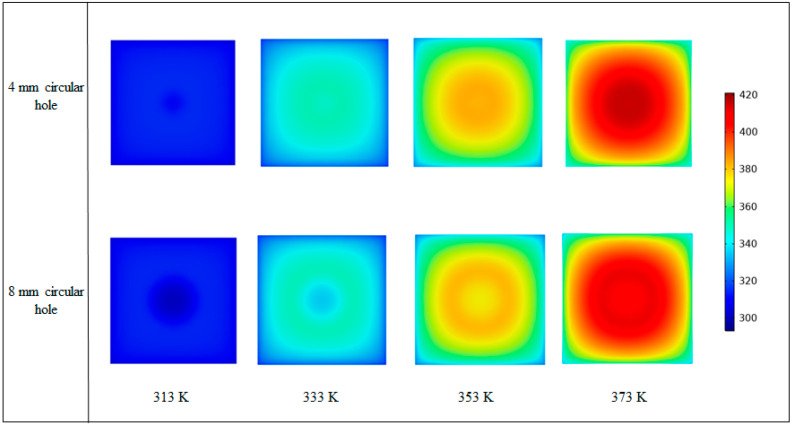
Temperature distributions of the insert with transition layers containing 4 and 8 mm circular holes.

**Figure 15 nanomaterials-12-01751-f015:**
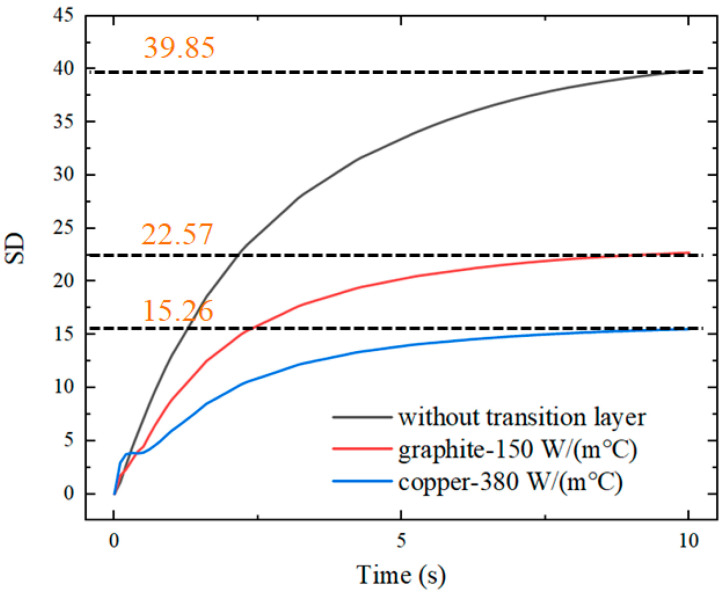
SD of the temperature of the insert with different transition layers.

**Figure 16 nanomaterials-12-01751-f016:**
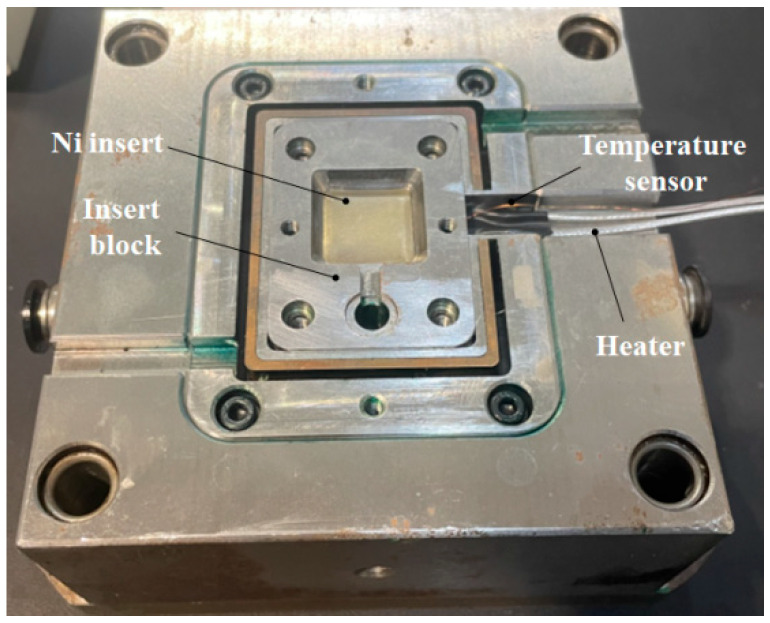
Developed mold with a variotherm system.

**Figure 17 nanomaterials-12-01751-f017:**
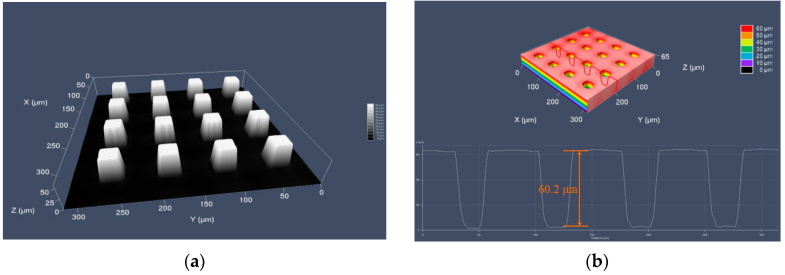
(**a**) The micropillars on a silicon wafer detected by laser confocal microscopy (LSM) and (**b**) microholes on a nickel insert detected by LSM.

**Figure 18 nanomaterials-12-01751-f018:**
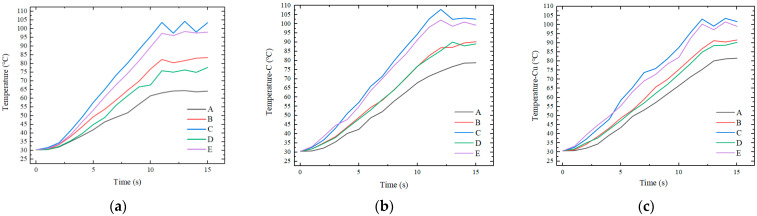
Temperatures at five sampling points: (**a**) without a transition layer, (**b**) with a graphite transition layer containing an 8 mm hole and (**c**) with a copper transition layer containing an 8 mm hole.

**Figure 19 nanomaterials-12-01751-f019:**
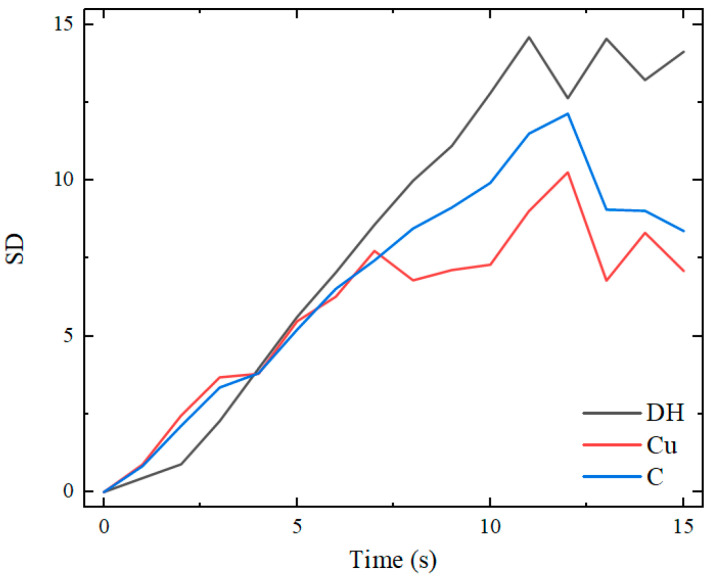
SD of the surface temperature of the insert under three heating modes.

**Figure 20 nanomaterials-12-01751-f020:**
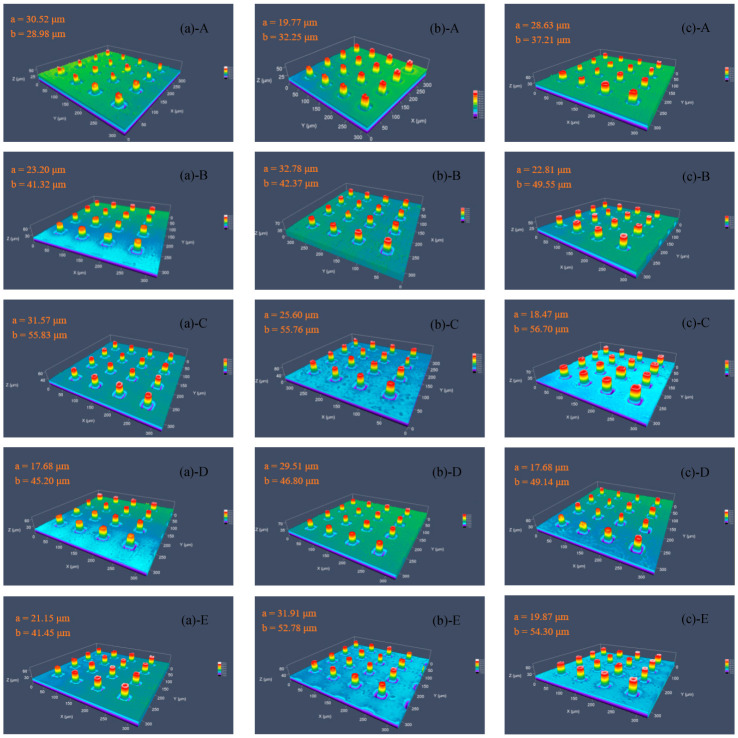
Replicated micropillars of five regions in different heating conditions, (**a**) without transition layer-direct heating, (**b**) graphite transition layer with 8 mm holes and (**c**) copper transition layer with 8 mm holes.

**Table 1 nanomaterials-12-01751-t001:** Material properties of the mold.

	20 °C	100 °C	200 °C	350 °C
λ (W/m°C)	34.5	34.2	33.9	33.5
C (J/kg°C)	500	550	600	625
ρ (kg/dm^3^)	7.75	7.75	7.75	7.75

## Data Availability

The data presented in this study are available on request from the corresponding author.

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
