# Peer review of "Application and Optimization of the Thin Electric Heater in Micro-Injection Mold for Micropillars"

_nanomaterials, 2022, doi:10.3390/nano12101751_

Round 1
Reviewer 1 Report
Please find the attached Review file.

Reviewer 2 Report
In the paper "Application and optimization of the thin electric heater in micro-injection mold for micropillars" the performance of the thin electric heater in a variotherm system is investigated by combining numerical simulations with experiments.
The research presented belongs to applied engineering. The topic of the research fits the journal's interest, and no major flows can be recognized during the lecture.
Despite these positive aspects, the quality of the paper is not ready for publication for the following reason:
1) English language must be improved. Especially in the Introduction, whose first 40 lines must be completely rewritten since the reader is not able to understand in the present form.
2) The device must be clearly explained in the text (more than what has been done)
3) The methods must be explained, specifically which kind of simulation has been done.
4) The results must be more clearly presented. Since most of the argumentation is based on comparisons, the diagram must clearly present them (include the comparison in the figure)
5) Do the experiments and the modeling really fit together?
6) In the title the Author wrote "application". There is no discussion about the applications. Is the title adequately clear?
Minor inaccuracy:
a) at line 46. Padua is in Italy, not in Ireland.
Round 2
Reviewer 1 Report
The paper has been substantially revised and improved. In general, the authors have managed to address my comments successfully. Therefore, I suggest accepting the paper.
Reviewer 2 Report
This contribution is a revised version of a previous submission.
The quality of the presentation is now high enough to be advised for publication